# Optimal Frontier-Based Autonomous Exploration in Unconstructed Environment Using RGB-D Sensor

**DOI:** 10.3390/s20226507

**Published:** 2020-11-14

**Authors:** Liang Lu, Carlos Redondo, Pascual Campoy

**Affiliations:** Centre for Automation and Robotics (C.A.R.), Computer Vision and Aerial Robotics Group (CVAR), Universidad Politécnica de Madrid (UPM-CSIC), Calle José Gutiérrez Abascal 2, 28006 Madrid, Spain; carlos.redondop@alumnos.upm.es (C.R.); pascual.campoy@upm.es (P.C.)

**Keywords:** aerial robot, autonomous exploration, information gain, RGB-D sensor

## Abstract

Aerial robots are widely used in search and rescue applications because of their small size and high maneuvering. However, designing an autonomous exploration algorithm is still a challenging and open task, because of the limited payload and computing resources on board UAVs. This paper presents an autonomous exploration algorithm for the aerial robots that shows several improvements for being used in the search and rescue tasks. First of all, an RGB-D sensor is used to receive information from the environment and the OctoMap divides the environment into obstacles, free and unknown spaces. Then, a clustering algorithm is used to filter the frontiers extracted from the OctoMap, and an information gain based cost function is applied to choose the optimal frontier. At last, the feasible path is given by A* path planner and a safe corridor generation algorithm. The proposed algorithm has been tested and compared with baseline algorithms in three different environments with the map resolutions of 0.2 m, and 0.3 m. The experimental results show that the proposed algorithm has a shorter exploration path and can save more exploration time when compared with the state of the art. The algorithm has also been validated in the real flight experiments.

## 1. Introduction

During the last decade, the research focus on Micro Aerial Vehicles (MAVs) for search and rescue tasks has considerably increased [1,2,3,4,5]. Because of the small size and high maneuvering, MAVs can perform search and rescue missions more efficiently. Considering the payload of MAVs, only limited sensors can be carried on-board. For this reason, choosing the type of sensors and building an efficient system for MAVs is still challenging in order to perform such autonomous missions.

In the application of the MAVs, a large number of works have been carried out in the area of autonomous exploration [6,7,8,9,10,11,12,13,14,15,16,17,18,19,20]. MAVs-based autonomous exploration needs miniaturized sensors which can provide rich information to successfully fulfill the endeavor. Particularly, LIDARs or cameras are the main sensors used by the research or industrial society which are able to provide high dimensional information for the autonomous exploration or inspection tasks. Compared with LIDARs, cameras such as RGB cameras, RGB-D cameras or event cameras have a huge advantage in the price and size.

In this paper, an RGB-D camera which can provide point clouds is used as the main sensor to perform the perception tasks. Then, the information from the RGB-D camera is used for the mapping and planning module to find a feasible path that has more environment information. The OctoMap for mapping and Information Gain (IG) is used to search the optimal goal position and orientation for exploration. With the optimal goal, a safe corridor-based A* algorithm is used to plan safety and feasible path. Finally, position control is applied to fly the path.

There are three main innovations in our work:

First of all, the efficent strategy for generation frontier points, filtering frontier points using the k-means++ and finding the optimal frontier using IG information. Secondly, combining a safe corridor-based A* planning algorithm with a position controller to move to the optimal frontier. At last, the algorithm is integrated with our aerial robot framework and tested in a custom edge computing aerial platform.

The remainder of the paper is organized as follow. Section 2 presents the related works. The robot architecture and environment definition are described, and the problem is formulated in Section 3. The proposed optimal frontiers generation and motion planning algorithm are explained in Section 4. Section 5 introduces the algorithm complexity. The experimental results and discussion are shown in Section 6 and Section 7, respectively, and finally, Section 8 concludes the paper and summarizes future research directions.

## 2. Related Works

A number of methods have been proposed during the last few decades addressing the problem of autonomous exploration in unconstructed environments. There are two fundamental types of approaches: the frontier-based approaches and sampling-based approaches. The frontier-based approaches try to maximize the exploration efficiency by selecting and exploring the optimal frontiers between the known and unknown area of a map, while the sampling-based approaches randomly generate robot states and search the path which can maximize the information gathered in the environment. There are also works [8,9] that combine the frontier-based and sampling-based approaches. In recent years, the learning-based strategies [10,11,21] such as reinforcement learning, are also used to solve the autonomous exploration problem. In this paper, the classic autonomous navigation algorithms are mainly discussed. In the remainder of the section, firstly the literature using the frontier-based approach is introduced and then the work using the sampling-based method are explained; finally, the research using the hybrid exploration strategy is summarized.

### 2.1. Frontier-Based Autonomous Exploration

The frontier-based exploration approach was first introduced by B. Yamauchi in the year of 1997 [12]. Considering the limited computing resources on-board MAV, S. Shen et al. [22] proposed an autonomous exploration algorithm only using the information of the known occupied space in the current map. After evaluating the information in the regions determined by the evolution of a stochastic differential equation, the ones of most significant particle expansion correlated to unexplored space and then MAV started to explore these unknown regions. In 2013, C. Dornhege and A. Kleiner proposed a frontier-void-based autonomous exploration approach [23]. They combined two concepts of voids, which are the unexplored volumes and frontiers. Their approach has been evaluated in a mobile platform with a five DOF manipulator. Recently, S. Bai et al. [13] used Bayesian optimization to predict mutual information (MI) of the frontiers and then trained a Gaussian process (GP) to estimate the MI in the robot’s action space and chose the next exploration step. C. Wang  et al. [14] proposed a multi-objective reward function that could minimize both the map entropy and the path cost during the exploration process, then the frontier that had the best reward would be chosen as the goal, finally, a potential field is construed to guide the robot to the goal. H. Umari et al. [15] presented a local and global RRT-based frontier detection for candidate frontiers generation and then a function with information gain and navigation cost is applied to select the optimal frontier. Because of the local and global RRT-based frontier generation, this approach has a high algorithm complexity. T. Cieslewski et al. [16] proposed a frontier selection method for high speed flight. They used a function to select the frontier that had the minimum change in the velocity to reach it. Their strategy could significantly reduce the exploration time because of the high exploration speed. The have been more works in the last two years, an effective exploration strategy using expected information gain is proposed by E. Palazzolo and C. Stachniss [24]. Their method could reduce the risk of collision, while still maximizing the information gain in the exploration process. C. Gomez et al. introduced a topological frontier-based exploration using semantic information [25]. They combined frontier-based concepts with behavior-based strategies in order to build a topological representation of the environment.

### 2.2. Sampling-Based Autonomous Exploration

As one of the most famous sampling-based exploration methods, Receding Horizon “Next-Best-View” Planner (RHC-NBVP) [17] was firstly proposed by A. Bircher et al. RHC-NBVP repeatedly expanded a rapidly-exploring random tree and selected the best next node to explore using an objective function that considered the set of visible and unmapped voxels from robot configuration. This approach had two main disadvantages: (1) keep repeatedly expanding RRT is computing expensive; (2) select the optimal next step in the field of view might lead to a sub-optimal solution. C. Papachristos et al. [18] presented an RHC-NBVP considering about the localization and mapping uncertainty. This implementation helped the planner run in challenging environments like dark or clutter environments. A two stage optimized next-view planning framework is developed by Z. Meng et al. [26]. The two-stage planner consists of a frontier-based boundary coverage planner and a planner returns an exploration path with the consideration of global optimality in the context of accumulated space information. Thinking about reducing the computing resources of the RHC-NBVP, C. Witting et al. proposed the history-aware strategy [19]. Their planner would maintain and use a history of visited places so that they can improve the sampling efficiency. The Visual Saliency–aware NBVP is introduced by T. Dang et al. [20]. They used visual attention to build the maps that are annotated regarding the visual importance and saliency of different objects and entities in the environment. Then the visual saliency information was used by the planner to find the next best step. In the year of 2020, L. Schmid et al. [27] presented an RRT*-inspired online informative path planning algorithm. This algorithm could achieve global coverage and maximize the utility of a path in a global context, using a single objective function.

By comparing with the algorithms above, a frontier-based exploration algorithm using global information gain that could avoid the sub-optimal path is used. In order to achieve good exploration, the robot that performs the exploration missions is moving at a low velocity.

## 3. Preliminary

In this section, firstly the architecture of the robot and the definition of the environment are given and then the problem is formulated.

### 3.1. Robot Architecture and Environment Definition

The robot used in this paper is an aerial robot that has six degrees of freedom. These six degrees of freedom are the translational movements in the X, Y, and Z axes and the rotational movements which are roll, pitch, and yaw. The onboard sensor of the robot is a visual–inertial sensor with an RGB-D camera that can be used to perform localization and realize the environment.

A 3D occupancy grid map generated by using OctoMap [28] is applied to describe the environment and the gathered data from the RGB-D camera are used to provide the point clouds for updating the map.

The 3D occupancy map is organized by voxels, and every voxel has one state which is free, occupied or unknown. With OctoMap, every voxel has an occupied possibility. The value of the occupied possibility is in the range of 0 to 1. For the voxel *x*, the entropy e(x) can be calculated using the occupied possibility p(x) as in Equation (Equation 1).
(1)e(x)=−(p(x))log(p(x))−(1−p(x))log(1−p(x))

The entropy can be used to describe the uncertainty of the voxel. Lower values of the entropy imply a lower value of the uncertainty. The sum of the voxels’ entropy can be used to guide the autonomous exploration to reduce the map uncertainty.

In this work, the voxels whose occupied possibility value are higher than 0.5 are thought as the obstacle voxels.

### 3.2. Problem Formulation

The goal of the autonomous exploration is to explore an unknown environment as fast as possible. To improve the efficiency of the exploration algorithm, the width, length, and height of the environment are previously defined in this work.

The width, length and height of the environment are assumed *w*, *l* and *h*, the volume of every voxel is $V_{voxel}$, the guide goal pose is $G_{i}\left(x_{i},y_{i},z_{i},\theta_{i}\right)$ and the start pose of the robot is $\left\{P_{0}\left(x_{0},y_{0},z_{0},\theta_{0}\right)|-w/2<x_{0}<w/2,-l/2<y_{0}<l/2,0<z_{0}<h\right\}$. The robot starts to move from $P_{0}$ to $G_{i}$. The exploration process ends when the algorithm cannot find a new $G_{i}$. If the sum of $V_{voxel}$ is equal to w×l×h, it is thought of as a successful running of the exploration algorithm. The exploration process can be seen in Figure 1.

## 4. Methodology

In this section, the optimal frontier selection approach and the motion planning and control strategy are explained in detail. Firstly the optimal frontiers generation approach, which generates the goal for the motion planning module, is described and then how we move the robot to the goal using the motion planning and control module is explained.

### 4.1. Optimal Frontiers Generation

The points between known and unknown area in the occupancy map are selected as frontier points. Then, the candidate points are generated around these frontier points. The candidate points are generated using the formula below.
(2)$$\begin{array}{cc} \; & R_{i+1}=R_{i}+\Delta R;\\ \; & \theta_{i+1}=\theta_{i}+\Delta \theta ;\\ \; & i\in 0,1,\ldots ,i,\ldots ,n-1,n;\\ \; & R_{min}\leq R_{i}\leq R_{max};\\ \; & 0\leq \theta_{i}\leq 2\pi ; \end{array}$$
(3)$$\begin{array}{cc} \; & \left(X_{c},Y_{c},Z_{c},\theta_{c}\right)\\ \; & =(X_{f}+R_{i}\times cos\left(\theta_{i}\right),Y_{f}+R_{i}\times sin\left(\theta_{i}\right),Z_{f},\theta_{i});\\ \; & i\in 0,1,\ldots ,i,\ldots ,n-1,n; \end{array}$$

In Equation (Equation 2), $\theta_{i}$ is the yaw angle generated by the candidates generation algorithm. $R_{min}$ and $R_{max}$ is the minimum and maximum distances from the frontiers to the candidates. ΔR and Δθ are predefined increments for updating $R_{i}$ and $\theta_{i}$. In this article, ΔR and Δθ are chosen as 0.5 m and 12${}^{{}^{\circ}}$. In Equation (Equation 3), $X_{c}$, $Y_{c}$, $Z_{c}$ and $\theta_{c}$ are the *x*, *y*, *z* coordinate and yaw angle of the candidates, respectively. $X_{f}$, $Y_{f}$ and $Z_{f}$ are the *x*, *y*, *z* coordinates of the frontiers.

The distance from the candidate point to the frontier points is set as $D_{cf}$ and the distance to the robot is set as $D_{cr}$. With the candidates generated, firstly the candidates which are in the unknown area are removed and then the candidates whose $D_{cf}$ and $D_{cr}$ are smaller than a predefined value are removed.

With the generated candidates, the Kmeans++ algorithm is used for candidates clustering. By using the Kmeans++ algorithm, the number of the candidates can be reduced which can speed up the optimal frontier selection process.

The information value of the candidates after clustering can be calculated by using the information gain cost function [29]. By using the information value, the candidates can be sorted. The information gain cost function can be seen from Equation (Equation 4).
(4)$$G_{v}=\sum_{\forall r\in R_{v}}\sum_{\forall x\in X}e\left(x\right)$$

In Equation (Equation 4), $G_{v}$ is the information gain of the candidate points on the view *v*. $R_{v}$ is the set of rays cast through the candidate points on the view *v* in the robot’s field of view. *X* is the set of voxels each ray traverses through. e(x) is the mapping uncertainty of the voxel. We thought of the sum of e(x) of the candidate points on the view *v* as the information gain value.

After the candidates are sorted, they are sent to the motion planning and control module, the whole process of the optimal frontiers generation is given by Figure 2.

### 4.2. Motion Planning and Control

With the frontiers points from the optimal frontiers generation module, the motion planning module firstly selected the frontier point, which has the smallest value as the goal point.

Firstly, the A* path planning algorithm is used to plan a primitive path from the current point to the goal point. Then, the safe corridors are generated based on the path point. The safe corridor generation approach is similar, as in [30]. The occupancy grid including the path point is inflated from each of its six faces. The inflation of each face finishes when it reaches the obstacle voxel or the size of the inflated voxel reaches a predefined value. Finally, a path is replanned through the center of every safe corridor. By using this strategy, the replanned path is farther away from the obstacle than the previous path. The process is shown in Figure 3.

For the yaw planning, the yaw angle of the endpoint always looks towards the direction that has the largest IG gain while the other path points always look towards the next path point.

If the planner cannot find a feasible path to the goal or the planning time is higher than the predefined value, the current goal point are discarded and choose another frontier point with smaller value as the goal point.

After getting a feasible path, the planner sends the path points one by one to the motion controller. When there is a new obstacle detected in the path, the controller will stop the robot and let the planner find another feasible path. If there is no feasible path, the algorithm will generate candidates and search the feasible path again. The whole exploration process stops when the algorithm cannot find new frontiers. the whole process of the motion planning and control module is given by Figure 4.

## 5. Algorithm Complexity

The algorithm complexity is important when the computing resources of the agent are limited. For every run of the algorithm, firstly the frontiers are generated, then the information gain of the frontiers is computed and the best frontier with the most information is selected, and finally the A* algorithm is used to search the feasible path.
*Frontiers Generation*: the complexity of the frontier generation algorithm depends on the raw frontiers generation and Kmean++ clustser algorithm. As can be seen from Algorithm 1, The complexities of raw frontier algorithm and Kmean++ clustering algorithm are $O(2\pi \times \left(R_{max}-R_{min}\right)/\left(\Delta yaw\times \Delta R\right))$ and O(3×i×k×m), respectively. *i*, *k*, and *m* are the iteration times of the Kmean++ algorithm, the number of frontiers defined and the number of frontiers generated by raw frontiers generation, respectively.*Information Gain Computing*: The complexity of the gain estimation of each frontier point is the number of rays *n* in the FOV times the length of rays *l* to the obstacle divided by the resolution of the map *r*, which is O(n×l/r).*Path Search*: when using A* algorithm searching the feasible path in a 3D occupancy map, we assume A* algorithm can search six directions when expanding a new node and the number of nodes in the path is *a*. The complexity of the A* search is O(6×a). For the complexity of the safe corridor generation algorithm, it can be calculated by the minimum distance $L_{j}$ to the obstacle in the x,y,andz directions of every path node times the number of nodes *a* divided by the resolution of the map *r*, which is $O(a\times \sum_{j=0}^{a}L_{j}/r).$*Total Computational Complexity*: The total computational complexity per run of the algorithm is $O(2\pi \times \left(R_{max}-R_{min}\right)/\left(\Delta yaw\times \Delta R\right)+3\times i\times k\times m+n\times l/r+a\times \sum_{j=0}^{a}L_{j}/r).$

**Algorithm 1** Autonomous Exploration Path Generation.
**Input:** Map, $P_{init}$
**Output:** Path
1:$\left\{F_{0},F_{1},\ldots ,F_{m}\right\}$←$ExtraFrontier(P_{init},Map)$2:*n*← 03:
**for**
n<m
**do**
4:    yaw← 05:    **for**
yaw<2π
**do**6:        *R*←$R_{min}$7:        **for**
$R<R_{max}$
**do**8:           $F_{n}\left(x\right)$←$F_{n}\left(x\right)-R\cos\left(yaw\right)$9:           $F_{n}\left(y\right)$←$F_{n}\left(y\right)-R\sin\left(yaw\right)$10:           $F_{n}\left(z\right)$←$F_{n}\left(z\right)$11:           $F_{n}\left(yaw\right)$←yaw12:           **if**
$F_{n}\in unknown$
**then**13:               $F_{group}\left(n\right)$←$F_{n}$14:           **end if**15:           *R*←R+ΔR16:        **end for**17:        yaw←yaw+Δyaw18:    **end for**19:    *n*←n+120:
**end for**
21:$\left\{F_{0},F_{1},\ldots ,F_{k}\right\}$←$KmeanCluster(F_{group},k)$22:$IG_{max}$←0;*n*←0;opt← 023:
**for**
n<k
**do**
24:    $IG_{F_{n}}$←$IG(F_{n})$25:    **if**
$IG_{F_{n}}>IG_{max}$
**then**26:        $IG_{max}$←$IG_{F_{n}}$27:        opt←*n*28:    **end if**29:    *n*←n+130:
**end for**
31:Path←$AstarSearch(P_{init},F_{opt})$32:
**while**
$P_{i}\in Path$
**do**
33:    **while**
$P_{i}$
is
not
in
collision
**do**34:        $P_{new}\leftarrow Inflate\left(P_{i}\left(x\right),P_{i}\left(y\right),P_{i}\left(z\right)\right)$35:        **if**
$P_{i}$
is
in
collision
**then**36:           Replace
$P_{i}$
with
$P_{new}$37:           Break
While38:        **end if**39:    **end while**40:
**end while**



The process of the whole exploration algorithm is shown in Algorithm 1. The autonomous exploration path generation algorithm can be divided into three parts. These three parts are frontiers generation parts (from line 1 to line 20), optimal frontiers selection part (from line 21 to line 30), corridor-based A* path planning part (from line 31 to line 40). The detailed descriptions about these three parts can be seen in the Methodology.

## 6. Experiments and Results

### 6.1. Experimental Setup

For the simulation experiments, the experiments using the Robot Operating System (ROS) [31] are run on top of Ubuntu 18.04. The Gazebo simulator is used to provide the environment model and the RotorS simulation [32] is used to provide the physical model parameters of robot. All the experiments run on a laptop with Intel Core i7-8750H at 2.2 GHz. The robot in the simulation is the AscTec firefly, which is equipped with a RGB-D camera with a horizontal field of view of 135${}^{{}^{\circ}}$ and vertical field of view of 60${}^{{}^{\circ}}$. The sensing range of the RGB-D camera is set as 5 m. The camera has an angle of 6${}^{{}^{\circ}}$ looking down to the ground. In the simulation experiments, the ground truth is used to perform the robot localization and the robot starts in the origin with yaw angle of zero.

To evaluate our exploration algorithm, it is tested in three different environments. These environments are a small-size environment, a medium-size environment, and a big-size environment. They are named as the apartment environment, the maze environment, and the office environment. These three environments can be seen in Figure 5. The size of the flat environment is 15 m × 14 m × 3 m, the size of the maze environment is 20 m × 20 m × 2.5 m. The height of the office environment is 2.5 m and the area of it is 800 m${}^{2}$. Our algorithm is compared with two different algorithms. One algorithm uses Bayesian optimization and the Gaussian process to predict the mutual information for frontier selection [13] and the other algorithm uses the Euclidean distance to choose the nearest frontier to explore. The main differences between our algorithm and these two algorithms are the kmeans++ for candidate clustering and using information gain for calculating the information value of the frontiers. The algorithms are also tested in the same environment with resolutions.

The platform for the experiment is that shown in Figure 6. As can be seen from the figure, the aerial platform is the Parrot Bebop 2, the on-board computer is the Jetson TX2 4GB module with orbit carrier and the RGB-D sensor is the Intel Realsense D435i, respectively. The depth camera D435i is part of the Intel RealSense D400 series of cameras, a lineup that takes Intel’s latest depth-sensing hardware and software and packages them into easy-to-integrate products. Adding an IMU allows the application to refine its depth awareness in any situation where the camera moves. This opens the door for rudimentary SLAM and tracking applications allowing better point-cloud alignment. The minimum depth distance of the depth camera is 0.0105 m while the horizontal and vertical FOVs of are 86 and 57${}^{{}^{\circ}}$, respectively. The whole syste, including the controller, mapping and exploration module is run on the Jetson TX2 with a Dual-core NVIDIA Denver 2 64-bit CPU and NVIDIA Pascal™ Architecture GPU with 256 CUDA cores. The exploration algorithm is integrated with the Aerostack frame work [33] in the real flight. The system architecture used in this work is shown in Figure 7.

As can be seen in Figure 7, there are five main parts in the system architecture that are human operator, execution module, communication module, sensor system, and actuator system. For the part of the human operator, three basic commands are used. These three basic commands are *take off*, *position control* and *land*. The basic commands can be sent to the onboard computer via Wifi. The other four parts including the autonomous exploration algorithm run from the onboard computer and the outputs are the roll, pitch, yaw, and thrust commands. The control commands are sent to Parrot Bebop 2 via a USB cable that connect the onboard computer and drone.

### 6.2. Simulation Experiments

The simulation results of the exploration path in the office, maze, and flat environments are shown in Figure 8. The start point of the exploration algorithms in the three environments with different mapping resolutions are all (0,0,1) m.

We limit the maximum exploration time of the office, maze, and flat environments as 1500 s, 1200 s, and 500 s. In this article, BOGP and NF are used to express the Bayesian optimization and the Gaussian process-based algorithm and the nearest frontier-based exploration algorithm. Each algorithm is run five times in the flat, maze and office environment with different resolutions. Table 1 shows the exploration time of these three algorithm under different map resolutions. The results of BOGP-based algorithm in the office environment with the map resolution of 0.2 m are not included because the exploration time is far more than the limited time.

### 6.3. Real Flight Experiments

In order to validate the proposed algorithm, a real flight experiment is given that runs the algorithm on-board an MAV. The experiment was conducted in a 4 m × 2 m × 2 m scenario equipped with a Optitrack motion capture system providing the MAV pose. Figure 9 shows the scenario for the MAV to navigate and the mapping for visualization. As can be seen from the figure, the cylider is put as an obstacle and the MAV will navigate around it. The cylinder obstacle is put at the position of (1.3,−0.1)/m. The height and radius of the cylinder are 1.26 m and 0.1 m, respectively. The boxes and board show the boundary of the scenario. The exploration time and the length of the exploration path is 44.9 s and 9.0398 m. P1 and P2 is the selected optimal frontier points. As can be seen from Figure 10, the robot first flew to P1 and then to P2. When the robot arrived P2, the final optimal frontier point P3 was selected. The exploration process was finished when robot arrived P3.

A video description about the experiments is shown at the following link: https://vimeo.com/455285058/f969e81451.

## 7. Discussion

As can be seen from Figure 8, if the exploration time is not limited, all the exploration algorithms can explore the whole environment when the map resolution is 0.2 m and 0.3 m.

The performance improvement of the proposed algorithm can be more clear when the exploration time is limited. As shown in Figure 11a,c,e, when the resolution of the map is 0.2 m (the solid line in the figure), the proposed algorithm (red line) can finish the exploration tasks in the limited time in all the scenarios. For BOGP based and NF based algorithm, it can finish the exploration tasks but takes a longer time than the proposed algorithm. In Figure 11a, the BOGP-based exploration cannot finish the exploration in the limited time when the map resolution is 0.2 m (the coverage of the whole scenario is 2000 m${}^{3}$, while it only reaches around 1250 m${}^{3}$). It is because the Bayesian optimization and Gaussian processes take a long time to analyze the map and this becomes more clear when performing in a big environment like the office scenario. As can be seen from Table 1, when the map resolution is 0.2 m, the proposed algorithm takes 255 s, 808 s, and 1495 s on average to finish exploration, while BOGP-based and NF-based algorithms take a much longer time. When the map resolution is increased to 0.3 m, as shown in Figure 11a,c,e, and Table 1, the performance of NF based algorithm is close to the proposed algorithm but the proposed algorithm still can finish the task faster.

Figure 11b,d,f show the length of the exploration path changing during the exploration process. It can be seen in Figure 11d,f and Table 1 that the proposed algorithm has a shorter exploration path than the NF-based algorithm in all the three scenarios and has a shorter path than the BOGP-based algorithm in the maze scenario with both resolutions and flat scenario with the resolution of 0.2 m. The results show that both optimal frontier selecting functions of the proposed and BOGP-based algorithm can help the exploration algorithm follow a shorter path for fulfilling the exploration task. In the office environment, with a resolution of 0.2 m, our algorithm has a longer exploration path length because the BOGP-based method only explores around half of the whole environment in the limited time.

Figure 11 shows one run for each method with different resolutions. Repeated experiments confirm that these curves are representative.

## 8. Conclusions and Future Works

In this article, an autonomous exploration strategy using the information gain to select the optimal frontier is presented. Firstly, an RGB-D camera to sense the environment and OctoMap is used to express the obstacle, free, and unknown space in the environment. Secondly, the Kmean++ algorithm is used to filter the frontiers and an information gain cost function is applied to select the optimal frontier to explore. Finally, A* path planner is applied to find the path and a safe corridor generation algorithm is used to move the path point far away from the obstacle before sending the path to the controller. The proposed algorithm has been compared with BOGP- and NF-based algorithms in three different environments with different map resolutions, in which the experimental results show that the proposed algorithm can save more exploration time than the other two algorithms and it also has a shorter exploration path.

The future work will focus on reducing the algorithm complexity and achieve higher exploration efficiency. A reinforcement learning-based exploration approach will also be considered in the future. The proposed exploration algorithm will also be implemented with Visual Inertial Odometry to perfrom the large environment exploration.

## Figures and Tables

**Figure 1 sensors-20-06507-f001:**
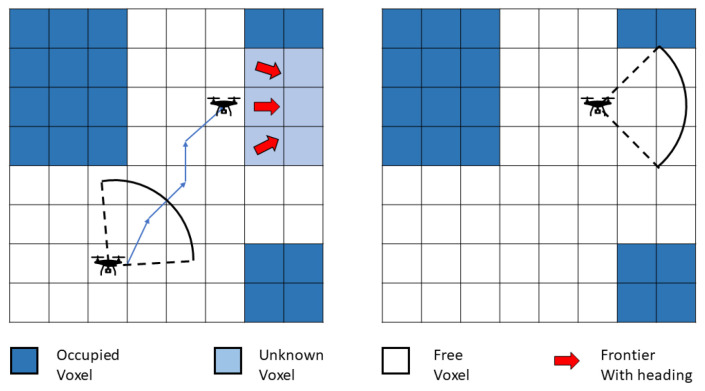
The schematic diagram for the exploration process. The robot starts exploration by moving to the selected frontier and finishes exploration when there is no new frontier found.

**Figure 2 sensors-20-06507-f002:**
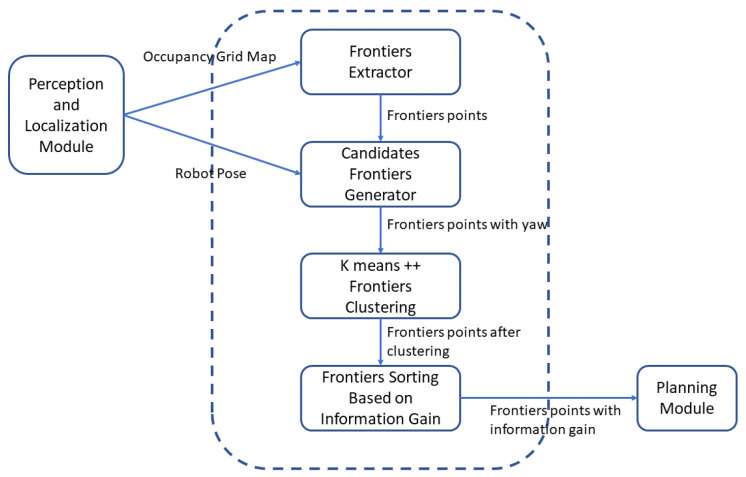
The flow chart for the optimal frontier generation process.

**Figure 3 sensors-20-06507-f003:**
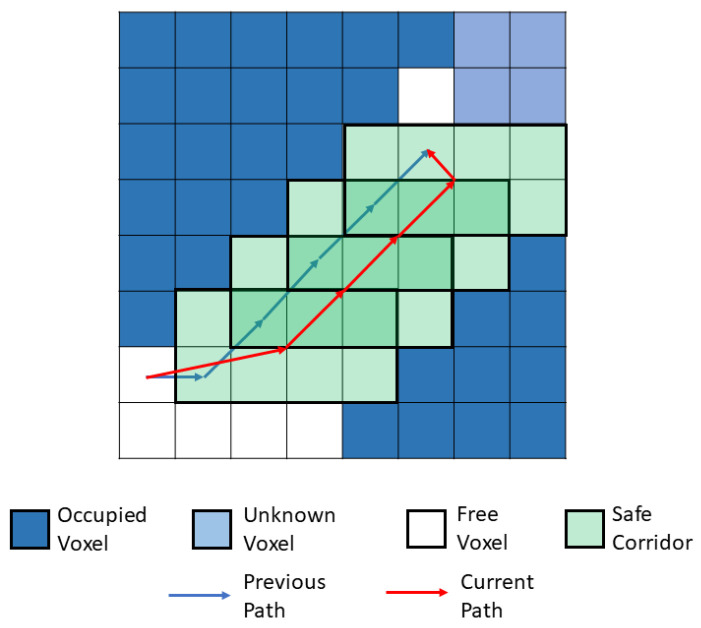
The schematic diagram for safe corridor generation process.

**Figure 4 sensors-20-06507-f004:**
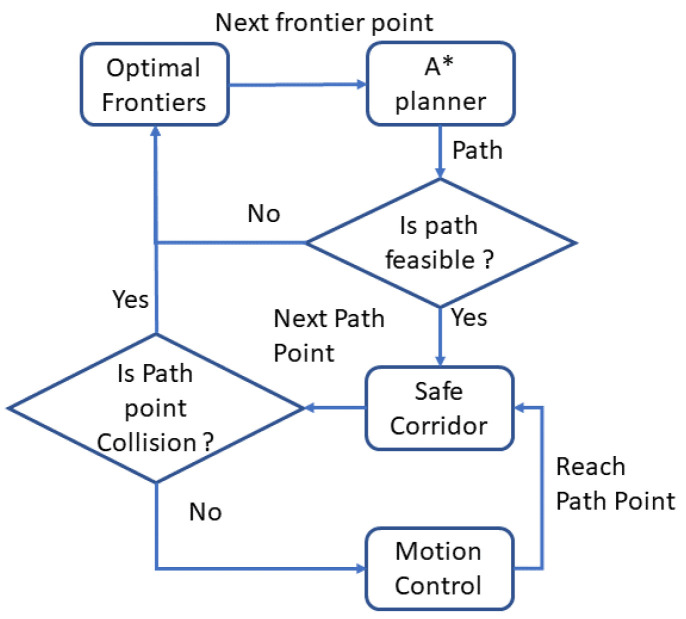
The flow chart for the motion planning and control process.

**Figure 5 sensors-20-06507-f005:**
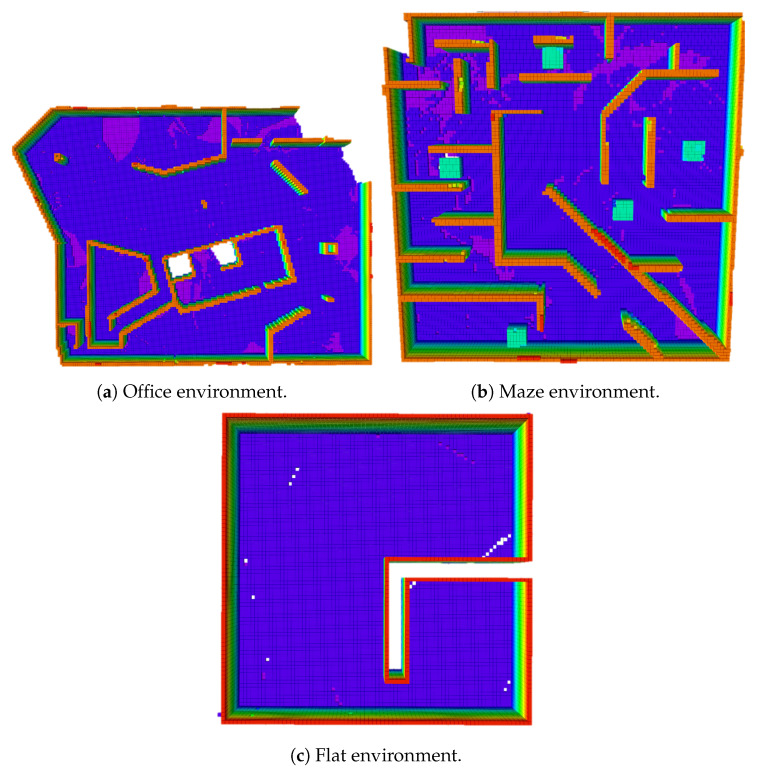
Three simulation environments for testing the exploration algorithm.

**Figure 6 sensors-20-06507-f006:**
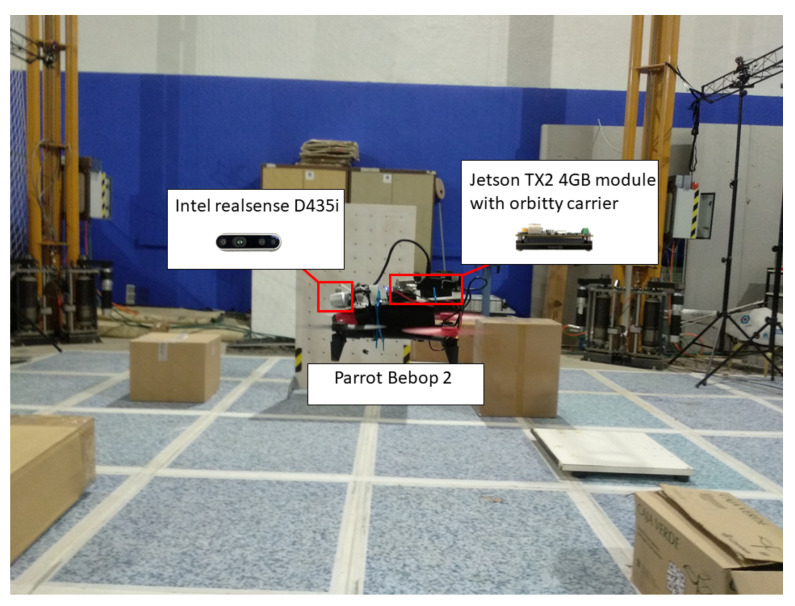
Aerial platform used in the real flight experiment.

**Figure 7 sensors-20-06507-f007:**
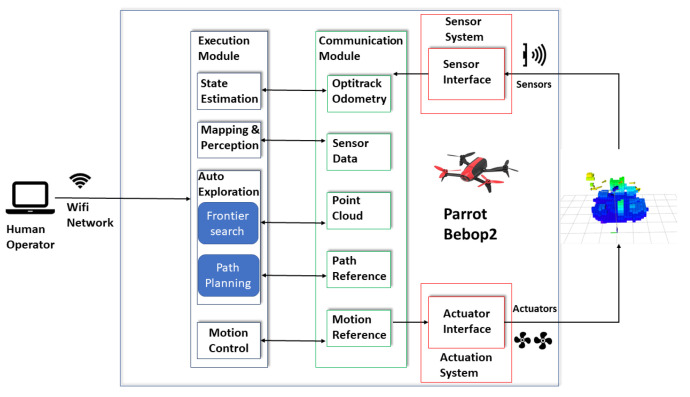
The system architecture used in the real flight experiment (based on the Aerostack framework). The optitrack odometry is provided by the optitrack motion capture system. The position and steering tracking error of it is less than 0.03 mm and 0.05${}^{{}^{\circ}}$, respectively.

**Figure 8 sensors-20-06507-f008:**
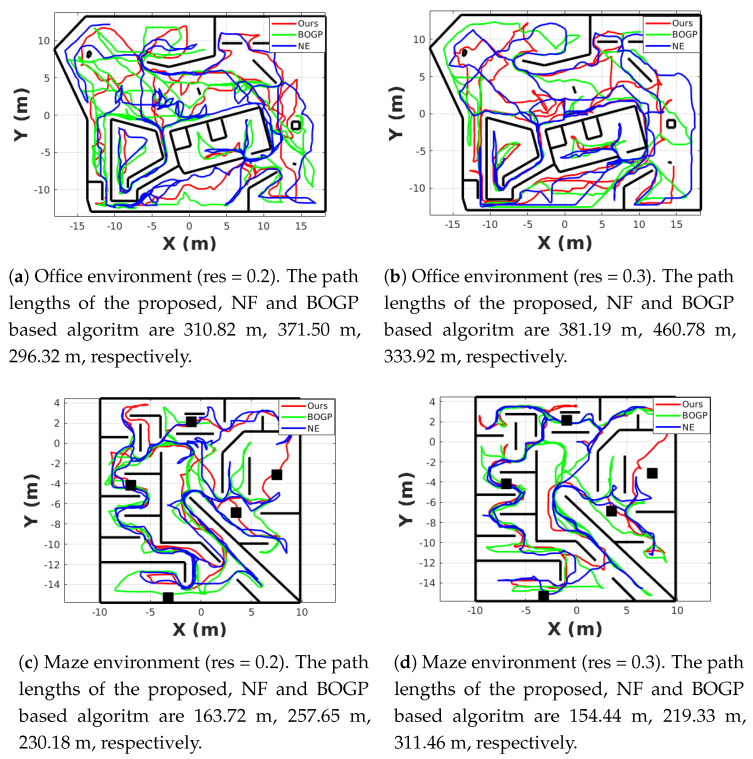
The simulation results of the exploration path of one run. The red line, green line, and blue line are our algorithm, the algorithm using Bayesian optimization and Gaussian process and the algorithm using the nearest frontier, respectively.

**Figure 9 sensors-20-06507-f009:**
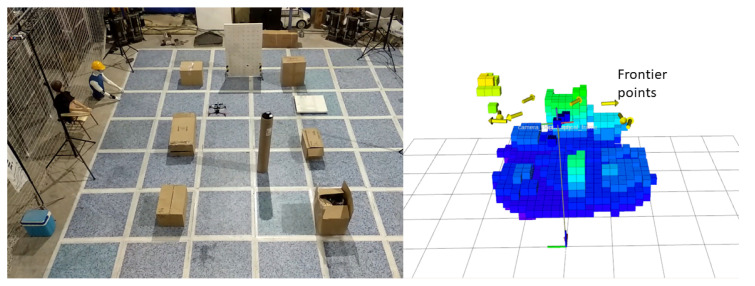
(**Left**) A photograph of the MAV and the scenario. (**Right**) Map of the scenario the experiment was conducted in. Voxels are coloured based on their height and the objects have been cropped for visualization.

**Figure 10 sensors-20-06507-f010:**
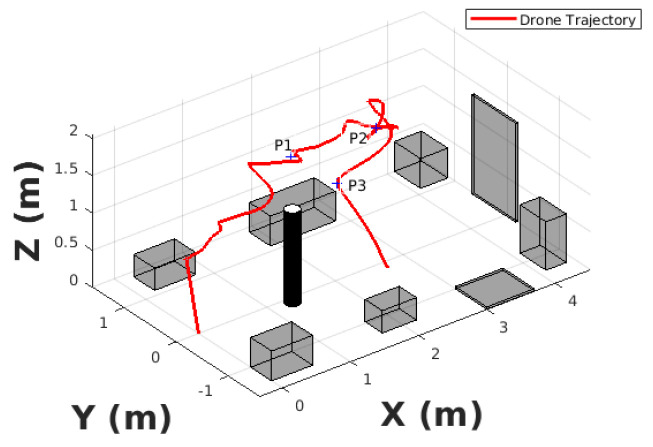
The results of the real flight experiments. The red line is the trajectory of the aerial robot.

**Figure 11 sensors-20-06507-f011:**
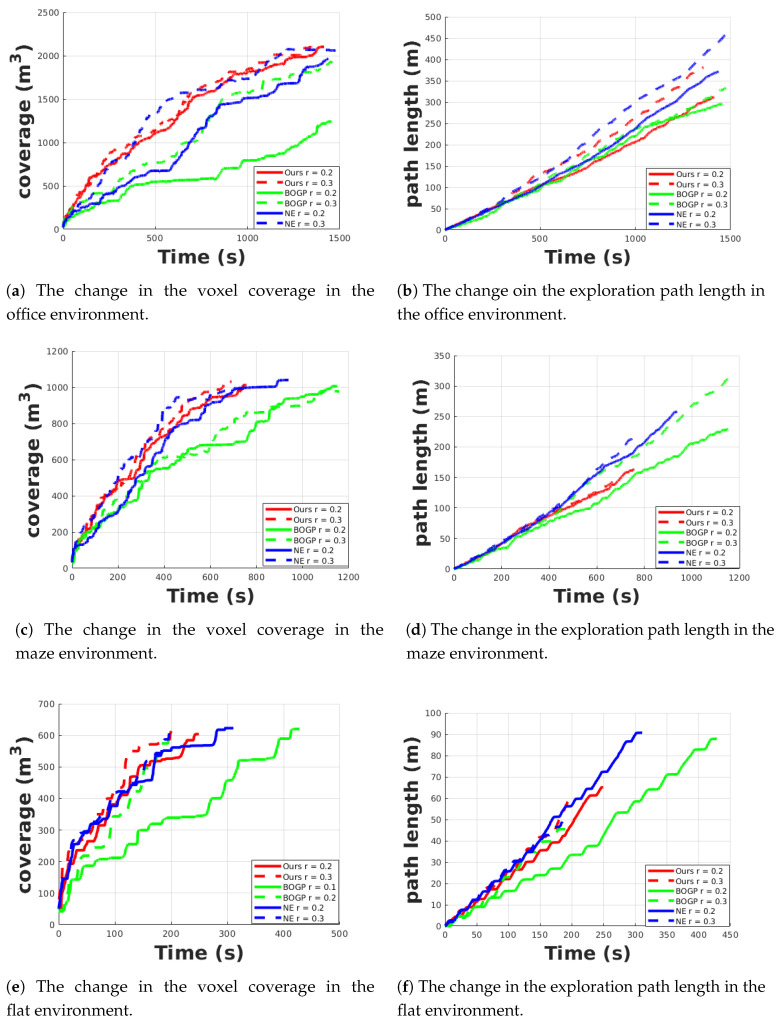
One run of the changing of the voxel coverage and exploration path in the office, maze, and flat environment. the red line, green line, and blue line are the results of the algorithm using Bayesian optimization and gaussian process and the algorithm using the nearest frontier, respectively. The solid line and the dashed line mean the resolution of mapping is 0.2 m and 0.3 m.

**Table 1 sensors-20-06507-t001:** Comparison of the proposed algorithm and baseline algorithm. The bold number in the table means the best performance in the comparison of each group in the table.

Environment	Map Resolution (m)	Exploration Algorithm	Exploration Time (s)	Path Length (m)
flat env	0.2	Ours	**255 ± 9.12**	**75.56 ± 14.76**
		NF	326 ± 31.21	89.37 ± 1.93
		BOGP	411 ± 23.33	92.43 ± 6.46
	0.3	Ours	**209 ± 12.72**	63.16 ± 1.19
		NF	227 ± 43.84	67.15 ± 8.27
		BOGP	217 ± 23.33	**52.48 ± 4.97**
maze env	0.2	Ours	**808 ± 72.12**	**192.26 ± 40.36**
		NF	905 ± 46.67	271.72 ± 19.90
		BOGP	1082 ± 98.99	228.34 ± 2.60
	0.3	*Ours*	**650 ± 59.3**	**180.27 ± 35.27**
		NF	761 ± 23.33	223.11 ± 19.50
		BOGP	1051 ± 151.32	306.18 ± 7.47
office env	0.2	Ours	**1495 ± 75.57**	**355.58 ± 30.29**
		NF	1609 ± 141.02	366.50 ± 6.41
		BOGP	-	-
	0.3	Ours	**1361 ± 81.63**	379.50 ± 7.42
		NF	1515 ± 113.61	449.06 ± 10.48
		BOGP	1576 ± 207.01	**374.41 ± 57.26**

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
