# Peer review of "Optimal Frontier-Based Autonomous Exploration in Unconstructed Environment Using RGB-D Sensor"

_sensors, 2020, doi:10.3390/s20226507_

Round 1
Reviewer 1 Report
I have gone through the paper entitled" Optimal Frontier Based Autonomous Exploration in Unconstructed Environment Using RGB-D Sensor". This paper presents an autonomous exploration algorithm for the aerial robots that shows several improvements for being used in the search and rescue tasks. The strategy uses proposed algorithm has shorter exploration path and can save more exploration time when compared with the state of art. The innovation contribution of this method is not only to combine the safety corridor-based A* planning algorithm with a position controller to move to the optimal frontier, but also to integrate with the aerial robot framework. The structure of the article is complete and reasonable, and the diagrams and algorithms are clear. This approach is very interesting and I enjoyed reading the paper.
However, there are some unclear points as follows:
First, in this paper, an autonomous exploration algorithm is presents and validated in the real flight experiments. An RGB-D sensor is used to receive information from the environment, and a clustering algorithm and an information gain are applied to choose the optimal frontier, and last the feasible path is given by A* path planner and a safe corridor generation algorithm. I want to know what A* means in the article.
Second, in this paper, the author proposes that the future work should focus on reducing the algorithm complexity and improving the exploration efficiency. I want to know how to reduce the algorithm complexity.
Third, in order to highlight the advantages of this method compared with other similar methods, it is suggested that the optimization results will be listed in tabular form.
Author Response
Responses to Reviewer
P1. First, in this paper, an autonomous exploration algorithm is presented and validated in real flight experiments. An RGB-D sensor is used to receive information from the environment, and a clustering algorithm and an information gain are applied to choose the optimal frontier, and last, the feasible path is given by A* path planner and a safe corridor generation algorithm. I want to know what A* means in the article.
A1. In this article, A* is used as the path planning algorithm to plan a collision-free path from the current position of the robot to the optimal frontier point. The definition of the A* algorithm is listed below:
A* (pronounced "A-star") is a graph traversal and path search algorithm, which is often used in many fields of computer science due to its completeness, optimality, and optimal efficiency. One major practical drawback is its O(b^d) space complexity, as it stores all generated nodes in memory. Thus, in practical travel-routing systems, it is generally outperformed by algorithms that can pre-process the graph to attain better performance, as well as memory-bounded approaches; however, A* is still the best solution in many cases.
P2. Second, in this paper, the author proposes that future work should focus on reducing the algorithm complexity and improving exploration efficiency. I want to know how to reduce the algorithm complexity.
A2. The proposed algorithm used the greedy frontiers generating approach (as can be seen from line 141 to line 152 of this article) to generate the frontier points candidates. The complexity of the algorithm can be reduced using the optimal yaw approach that can find an optimal yaw angle for every candidate points instead of generating (2*pi/delta_yaw) candidates with different yaw angles. The yaw angles can be optimized by performing a sparse 360 degrees ray casting which at the same time computes the map entropy along each ray. The one with the highest cumulative map entropy is selected as the yaw angle for the candidate frontier points. The complexity of the frontier generation part of the algorithm would be reduced to O((R_max−R_min)/∆R + 3×i×k×m) by applying the above strategy.
P3. Third, in order to highlight the advantages of this method compared with other similar methods, it is suggested that the optimization results will be listed in tabular form.
A3. Thanks for this suggestion, the optimization results of the proposed and
benchmarking approaches are listed in Table.1.

Reviewer 2 Report
The paper is well written and includes a nice set of experiments and practically oriented algorithms to reduce path lengths and exploration times on an autonomous quadcopter.
The paper is believed to be acceptable for publication, but some changes are suggested:
1- To use consistent notation in algorithm 1: The arrows are used for incrementation on some lines (eg. Fn(x) incrementation) while equality symbol is used on others (eg. R=R+DeltaR)
Use of English: Some changes are needed throughout the paper, for example:
2- Remove the C capital letter in Secondly, combining (second to last pragraph of the introduction.
3- Add the word "that" between robot and performs "The robot that performs"
4- Before equation 1: Replace "are" by "is" in "The 3D occupancy map are"
5- "The less the value of the entropy means the less" should read "Lower values of the entropy imply a lower" ...
6- In 3.2: "The exploration process finish" should read "The exploration process ends"
Other changes:
7- In figure 8, 'path lengthes' should read 'path lengths' in all figure captions
8- Authors should provide a brief justification of their hardware chosen for real time implementation.
9- More detail should be provided about the RGB sensor and is it the same as the 'perception sensor', if so the same terminology should be used for the same variable.
10- Authors also need to explain how the pose is determined using the Optitrack visual odometry system. There is no detail on that. Optitrack performance specs are not provided too.
11- Is the focus of Astar on optimising distance or on maximizing the information gain cost function? The problem formulation is not sufficiently clear as an optimization problem . Is the information gain optimisation to determine frontiers and Astar to minimize distance? This should be clearer.
Author Response
Responses to Reviewer
P1. To use consistent notation in algorithm 1: The arrows are used for incrementation on some lines (eg. Fn(x) incrementation) while equality symbol is used on others (eg. R=R+DeltaR).
A1. Thanks for the suggestion. As can be seen from algorithm 1, all the equality symbols are changed to the arrows.
P2. Remove the C capital letter in Secondly, combining (second to the last paragraph of the introduction).
A2. As can be seen in line 36 of this article, this change has been implemented.
P3. Add the word "that" between robot and performs "The robot that performs".
A3. Thanks for pointing out it. The change has been implemented in line 108 of this article.
P4. Before equation 1: Replace "are" by "is" in "The 3D occupancy map are".
A4. The modification has been done in line 119 of this article.
P5. "The less the value of the entropy means the less" should read "Lower values of the entropy imply a lower" ...
A5. Thanks for the suggestion. The change has been done in line 120 of this article.
P6. In 3.2: "The exploration process finish" should read "The exploration process ends".
A6. it has been implemented in line 132 of this article.
P7. In figure 8, 'path lengthes' should read 'path lengths' in all figure captions.
A7. done.
P8. Authors should provide a brief justification of their hardware chosen for real-time implementation.
A8. The hardware chosen for the proposed system is the same as [1]. As can be seen from [1], the hardware chosen can be run in real-time. [2] shows the real-time performance combining the Jetson TX2 with the Intel Realsense D435i RGB-D camera. The real flight experiments of this article also show that all the algorithms can be run in the Jetson TX2 in real-time.
P9. More detail should be provided about the RGB sensor and is it the same as the 'perception sensor' if so the same terminology should be used for the same variable.
A9. The RGB-D camera is the perception sensor in this article, and the “perception sensor” has already been changed to “RGB-D camera”. The RGB-D camera used in the real flight experiments of this article is the Intel Realsense D435i. The depth camera D435i is part of the Intel RealSense D400 series of cameras, a line-up that takes Intel’s latest depth‐sensing hardware and software and packages them into easy‐to‐integrate products. Adding an IMU allows the application to refine its depth awareness in any situation where the camera moves. This opens the door for rudimentary SLAM and tracking applications allowing better point-cloud alignment. The minimum depth distance is 0.0105 m. The horizontal and vertical FOVs of the depth camera are 86 and 57 degrees, respectively. The description of the RGB-D camera has been added in line 235 to line 240 of this article.
P10. Authors also need to explain how the pose is determined using the Optitrack visual odometry system. There is no detail on that. Optitrack performance specs are not provided too.
A10. The Optitrack is the motion capture system that can provide a highly accurate 6 DOF pose of the robot using tracking cameras. The position and steering tracking error of the Optitrack motion capture system is less than 0.03 mm and 0.05 degrees, respectively. More detailed information on the Optitrack can be found in [3]. The explanation of the Optitrack odometry has been added in the caption of Fig.7.
P11. Is the focus of Astar on optimising distance or on maximizing the information gain cost function? The problem formulation is not sufficiently clear as an optimization problem. Is the information gain optimisation to determine frontiers and Astar to minimize distance? This should be clearer.
A11. Thanks for the question. This Astar is focusing on optimizing distance and the information gain cost function is optimized by optimal frontier selection and Kmean++ cluster algorithm. The information gain is used to choose the optimal frontier and the Astar path planner is used to find the shortest path to reach it.
1.J. Lin, H. Zhu and J. Alonso-Mora, Robust Vision-based Obstacle Avoidance for Micro Aerial Vehicles in Dynamic Environments, 2020 IEEE International Conference on Robotics and Automation (ICRA), Paris, France, 2020, pp. 2682-2688.
2.Janus P., Kryjak T., Gorgon M. (2020) Foreground Object Segmentation in RGB–D Data Implemented on GPU. In: Bartoszewicz A., Kabziński J., Kacprzyk J. (eds) Advanced, Contemporary Control. Advances in Intelligent Systems and Computing, vol 1196. Springer, Cham.
3.Furtado J.S., Liu H.H.T., Lai G., Lacheray H., Desouza-Coelho J. (2019) Comparative Analysis of OptiTrack Motion Capture Systems. In Janabi-Sharifi F., Melek W. (eds) Advances in Motion Sensing and Control for Robotic Applications. Lecture Notes in Mechanical Engineering. Springer, Cham.
